# Correlation between Antimicrobial Resistance and the Hospital-Wide Diverse Use of Broad-Spectrum Antibiotics by the Antimicrobial Stewardship Program in Japan

**DOI:** 10.3390/pharmaceutics15020518

**Published:** 2023-02-03

**Authors:** Takashi Ueda, Yoshio Takesue, Kazuhiko Nakajima, Kaoru Ichiki, Kaori Ishikawa, Kumiko Yamada, Toshie Tsuchida, Naruhito Otani, Yoshiko Takahashi, Mika Ishihara, Shingo Takubo, Kosuke Iijima, Hiroki Ikeuchi, Motoi Uchino, Takeshi Kimura

**Affiliations:** 1Department of Infection Control and Prevention, Hyogo College of Medicine, Nishinomiya 663-8501, Hyogo, Japan; 2Department of Clinical Infectious Diseases, Tokoname City Hospital, Tokoname 479-8510, Aichi, Japan; 3Department of Public Health, Hyogo College of Medicine, Nishinomiya 663-8501, Hyogo, Japan; 4Department of Pharmacy, Hyogo College of Medicine Hospital, Nishinomiya 663-8501, Hyogo, Japan; 5Department of Clinical Technology, Hyogo College of Medicine, Nishinomiya 663-8501, Hyogo, Japan; 6Department of Inflammatory Bowel Disease, Hyogo College of Medicine, Nishinomiya 663-8501, Hyogo, Japan

**Keywords:** antibiotic stewardship program, antibiotic resistance, antibiotic diversity, *Klebsiella pneumoniae*

## Abstract

Increased antibiotic use and antibiotic homogeneity cause selective pressure. This study investigated the correlation between antibiotic diversity and antimicrobial resistance (AMR) in Gram-negative organisms. The days of therapy/100 patient-days (DOT) for four broad-spectrum antibiotic classes were evaluated for 2015–2022. The antibiotic heterogeneity index (AHI) for the equal use of four classes (25%) and the modified AHI for the equal use of three classes (30%), excluding fluoroquinolones (10%), were measured (target: 1.0). Quarterly antibiotic use markers and the resistance rates against ≥2 anti-*Pseudomonas* antibiotics were compared. The DOT value was 9.94, and the relative DOT were 34.8% for carbapenems, 32.1% for piperacillin/tazobactam, 24.3% for fourth generation cephalosporins/ceftazidime/aztreonam, and 8.9% for fluoroquinolones. Although no correlation was found between the total DOT and the resistance rate for any bacterium, a significant negative correlation was found between the heterogeneity indices and resistance rates for *Pseudomonas aeruginosa* and *Klebsiella pneumoniae*. The significant cutoffs that discriminate the risk of resistance were 0.756 for the AHI and 0.889 for the modified AHI for *K. pneumoniae*. Antibiotic diversity is more important in preventing AMR than overall antibiotic use. The ideal ratio of broad-spectrum antibiotics should be studied for diversified use to prevent AMR.

## 1. Introduction

There is a significant global burden attributed to bacterial antimicrobial resistance (AMR) including mortality, the duration of hospital stays, and healthcare costs [1,2]. Predictive statistical models have estimated that 4.95 million deaths are associated with AMR annually, and the six leading pathogens, namely *Escherichia coli*, *Staphylococcus aureus*, *Klebsiella pneumoniae*, *Streptococcus pneumoniae*, *Acinetobacter baumannii*, and *Pseudomonas aeruginosa* were responsible for 3.57 million deaths associated with AMR in 2019 [3]. Carbapenem-resistant Gram-negative bacteria, including carbapenem-resistant Enterobacteriaceae (CRE), *A*. *baumannii*, and *P*. *aeruginosa* cause difficult-to-treat infections and are associated with a poor prognosis [4]. Another concern regarding AMR in Gram-negative organisms is the community spread of extended spectrum β-lactamase (ESBL)-producing Enterobacteriaceae [5]. 

Urgent action is needed to prevent the increasing occurrence of resistant organisms and control their spread in hospitals. Prior use of antibiotics is the most significant risk factor for the acquisition of resistant Gram-negative organisms [6], and antibiotic stewardship (AS) programs have been advocated to reduce AMR-selective pressure [7,8,9,10]. Both overall antibiotic use (quantity assessment) and the diversity of antibiotic exposure (quality assessment) affect the antibiotic selective pressure [11,12,13,14,15,16,17,18]. Patients in tertiary care hospitals have higher overall antibiotic exposure than those in small community hospitals. However, a high prevalence of AMR organisms is not necessarily anticipated in tertiary care hospitals [19]. As an AS metric to assess the quantity of antibiotic use, the risk-adjusted antibiotic use metric known as the standardized antimicrobial administration ratio (SAAR) was developed [20]. Although strategies to achieve diverse antibiotic use have been suggested for intensive care units (ICUs) [12,21,22,23,24], few studies have reported AS metrics to evaluate hospital-wide antibiotic diversity [25]. 

Previously, we proposed the employment of the antibiotic heterogeneity index (AHI) for common broad-spectrum antibiotics with anti-*Pseudomonas* activity and observed that an increased AHI resulted in a decrease in the isolation rate of AMR organisms and improved antibiotic susceptibility in *P*. *aeruginosa* [25]. Although the equal use of broad-spectrum antibiotics, including carbapenems, piperacillin/tazobactam, fourth generation cephalosporins/ceftazidime, and fluoroquinolones, was suggested as a goal in terms of antibiotic diversity [21,22,23,24], no data have revealed whether this policy of equal use, which might promote the balanced use of antibiotics, is effective at decreasing selective pressure. Recent AS guidelines [26] recommended a prospective audit and feedback with timeouts, formulary restrictions and pre-authorization, education on appropriate antibiotic use, the preparation of practical manuals and clinical pathways, and the use of a computerized clinical decision-support system. However, no recommendation was made regarding a specific strategy aimed at diverse antibiotic use, such as antibiotic cycling or mixing [18,21,22,23]. Moreover, there was insufficient data regarding the degree of antibiotic diversity achieved by the AS program, which was aimed at appropriate antibiotic use in each patient. The purpose of this study was to evaluate the degree of antibiotic diversity achieved and to investigate the correlation between the diversity of antibiotic use and the isolation rate of AMR organisms using the 7 years of experience gained by the implementation of the AS program.

## 2. Materials and Methods

### 2.1. AS Programs

This study was performed at the Hyogo Medical University Hospital, which is a 963-bed tertiary care hospital. The study period was between April 2015 and March 2022. Quarterly data (from the first to the fourth quarter) in each year were collected, and the 7-year study period was divided into 28 terms. 

The study was approved by the Institutional Review Board of Hyogo Medical University (No. 2240). The AS team consisted of two infectious disease physicians, a pharmacist dedicated to the AS program, and an infection control nurse, with additional input from a microbiologist. Following the retirement of the leading AS physician, the other infectious disease physician involved in the study stepped up to lead the AS program in April 2021.

Prospective audits, feedback, and a consultation service with antibiotic timeouts are the main components of AS programs. Interventions were based on input from the microbiology laboratory (1, 2) or pharmaceutical department (3, 4) in the following situations: (1) when microorganisms were isolated from the aseptic specimen, (2) when multidrug-resistant organisms were isolated, (3) when piperacillin/tazobactam (since November 2017) or carbapenems (since September 2019) were initiated, and (4) when carbapenems or piperacillin/tazobactam were used for >10 days (>14 days for the remaining intravenous antimicrobial agents). AS rounds were conducted every weekday. Timeouts continued until the completion of antimicrobial therapy. The infectious disease physician or the pharmacist were available during working hours when the clinicians could consult them for advice regarding antibiotic therapy. 

### 2.2. Monitoring of Antibiotic Use

The total number of days of antibiotic therapy in a hospital during a specific timeframe divided by 100 bed days (which gives the days of therapy (DOT) value) was used to evaluate the amount of antibiotic use. The DOT with intravenous broad-spectrum antibiotics that were predominantly used for hospital-onset infections, such as carbapenems, piperacillin/tazobactam, fourth generation cephalosporins (i.e., cefepime)/ceftazidime/aztreonam, and fluoroquinolones, were measured. As a measure of the risk-adjusted antibiotic use in adult patients, the SAAR of broad-spectrum antibacterial agents predominantly used for hospital-onset infections was assessed in certain representative departments [20,27,28].

To assess the diverse use of these four antibiotic classes, the relative DOT (% DOT) and the AHI were evaluated. The % DOT was calculated as the DOT of one class of antibiotics divided by the DOT of all four classes of antibiotics. The AHI for the 25% equal use of these four classes was calculated as follows [14,25]:AHI = 1 − {n/(2 × [n − 1])} × Σ |ai − bi|
where n was 4 (number of antibiotic classes); ai was 0.25 for each antibiotic class; and bi was the proportion of the DOT for each antibiotic class. AHI was 1 when all antibiotic classes were equally used (% DOT = 25%). The modified AHI for the 30% equal use of three classes of antibiotics, except for the fluoroquinolones (10%), was also calculated by defining the ai as 0.1 for fluoroquinolones and 0.3 for other antibiotic classes. The abbreviation list is presented in Appendix A, and the method for the calculation of AHI and the modified AHI is summarized in Appendix A.

### 2.3. Monitoring of Antibiotic-Resistant Gram-Negative Rods

The evaluated organisms were glucose non-fermenting Gram-negative rods, including *P*. *aeruginosa*, *A*. *baumannii*, and Enterobacteriaceae species including *E*. *coli*, *K*. *pneumoniae*, *Klebsiella oxytoca*, *Klebsiella aerogenes*, *Enterobacter cloacae*, *Serratia marcescens*, *Citrobacter freundii*, *Proteus mirabillis*, *Proteus vulgaris*, and *Morganella morganii*. Stool specimens that were used for the surveillance cultures of AMR organisms were excluded. Organisms isolated from specimens obtained later than 48 h after admission were defined as nosocomial isolates. If ≥2 strains of the same bacterial species were isolated from a patient, the most resistant strain was chosen for the analysis. Bacterial identification and antimicrobial susceptibility testing were performed with MicroScan WalkAway (Beckman Coulter, Brea, CA, USA). Antibiotic susceptibility was determined according to the European committee on antimicrobial susceptibility testing (EUCAST) and the Clinical and Laboratory Standards Institute (CLSI) criteria [29] (Appendix A). As the primary endpoint for antibiotic susceptibility, resistance in EUCAST against ≥2 or ≥3 of the following antibiotic classes was analyzed: ciprofloxacin (levofloxacin for Enterobacteriaceae), cefepime, tazobactam/piperacillin, meropenem, or gentamycin/amikacin. The correlation between the markers of antibiotic use (AHI, modified AHI, or DOT) and the non-susceptibility rate against ≥2 or ≥3 classes of antibiotics was evaluated. 

Isolates of carbapenemase and ESBL-producing Gram-negative organisms were also compared. After screening based on cefazolin or ceftazidime/aztreonam minimum inhibitory concentration (MIC) criteria, the double disc synergy test was used to detect ESBL-producing Gram-negative bacteria [30]. Isolates with an MIC of: (1) ≥2 μg/mL to meropenem, (2) ≥2 μg/mL to imipenem and ≥64 μg/mL to cefmetazole, or (3) ≥16 μg/mL to ceftazidime and ≥32 μg/mL to cefoperazone/sulbactam, were screened for carbapenemase production. However, the screening criteria were changed to an MIC ≥0.5 μg/mL for meropenem or an inhibition zone diameter of <15 mm for faropenem in February 2020. The modified carbapenem-inactivation method [31], mercaptoacetic acid sodium salt (Eiken Chemical Co., Ltd., Tokyo, Japan), and 3-aminophenylboronic acid monohydrate (Kanto Chemical Co., Inc., Tokyo, Japan) [32] were used to detect carbapenemase-producing Gram-negative bacteria.

### 2.4. Statistical Analysis

Categorical variables were compared using the chi-squared or Fisher’s exact tests. Continuous variables are expressed as the mean (± standard deviation [SD]) and were compared using *t*-tests or Mann–Whitney U tests. Pearson’s correlation coefficient was calculated for the relationship between the rate of resistant organisms and antibiotic use markers (i.e., AHI, modified AHI, or DOT). The cutoff values were the maximum area under the curve (AUC) as determined through a receiver operating characteristic (ROC) curve. The level of statistical significance was set at *p* < 0.05. IBM SPSS Statistics for Windows version 24 (IBM Corp., Armonk, NY, USA) was used for all statistical analyses. 

## 3. Results

### 3.1. Intervention/Supervision of AS Programs

Because there was more than one episode at which AS could be initiated in certain patients, the number of events was evaluated instead of the number of patients. The mean annual number of events for AS was 1113.7 ± 89.2, and the mean annual cumulative number of interventions or supervisions for consultation, including timeouts, was 9688.4 ± 695.9. Thus, the number of interventions or supervisions required for one event was 8.71 ± 0.15. A summary of the patient background information for each year between April 2015 and March 2022 is presented in Appendix A. The annual number of admitted patients ranged from 22,820 to 24,415 by March 2020. However, the number of hospitalizations decreased to 21,931 in April 2020–March 2021 and 21,264 in April 2021–March 2022 because of the COVID-19 pandemic. The number of admitted COVID-19 patients was 49 in 2020, 164 in 2021, and 532 in 2022 (until March, 103). The rate of patients undergoing surgery (approximately 27–29%) and patient days in each department were comparable throughout the study.

The annual number of events according to the reasons for the intervention/supervision is presented in Table 1. The leading reason was consultation from physicians for the use of antimicrobial agents, followed by the suggestion of antimicrobial selection and dosing for patients admitted to the ICU. The consultation rates were 69.2% for April 2015–March 2016 and 65.0% for April 2016–May 2017. However, the rate decreased to 48.6%–56.2% after the initiation of the prospective audit and feedback for patients in whom piperacillin/tazobactam (November 2017) or carbapenems (September 2019) were used.

### 3.2. Antibiotic Use during the AS Programs

The DOT value was 9.94 for the consecutive 7-year study period. As a measure of risk-adjusted antibiotic use, the SAAR of adult broad-spectrum antibacterial agents predominantly used for hospital-onset infections in the Department of Respiratory Medicine, the Department of Colorectal Surgery, and the ICU is demonstrated in Appendix A. The % DOT values were 34.8% for carbapenems, 32.1% for piperacillin/tazobactam, 24.3% for fourth generation cephalosporins/ceftazidime/aztreonam, and 8.9% for fluoroquinolones administered intravenously. The AHI and modified AHI were calculated as 0.775 and 0.909, respectively. As 1.0 was the target for both indices, the modified AHI is a more feasible antibiotic heterogeneity marker to achieve a value closer to the target than the AHI in Japanese hospitals. The change in these antibiotic use markers for broad-spectrum antibiotics is demonstrated in Figure 1. In the earlier study period, the leading antibiotic was piperacillin/tazobactam. However, with the initiation of the prospective audit and feedback for piperacillin/tazobactam in November 2017, the % DOT of piperacillin/tazobactam gradually decreased. By contrast, the prospective audit and feedback for carbapenems that began in September 2019 did not influence the % DOT of carbapenems, which remained at approximately 40%. The % DOT of fluoroquinolones was approximately 10% throughout the study period.

In the evaluation of diverse antibiotic use, an AHI of >0.8 and a modified AHI of >0.9 were almost achieved between 3Q/2017 and 1Q/2019. However, a decrease in both antibiotic heterogeneity indices was observed from 4Q/2020 onwards, when more than 2000 new daily cases of COVID-19 were being reported for the first time in Japan. Although the total DOT for the evaluated broad-spectrum antibiotics gradually increased from 2Q/2015 (8.2) to 2Q/2017 (11.2), with the start of the prospective audit and feedback, the DOT decreased to 9.3 in 4Q/2017 and was sustained within the range of 9 to 10 until 2Q/2021. However, the DOT increased to approximately 11 after 2Q/2021. 

### 3.3. Isolation of Antibiotic-Resistant Gram-Negative Rods

The resistance rate against ≥2 antibiotic classes was 14.5% and that against ≥3 antibiotic classes was 7.2% in glucose non-fermenting Gram-negative rods and 14.1% and 3.8% in Enterobacteriaceae spp., respectively (Table 2). The rate of ESBL-producing Enterobacteriaceae spp. was 11.4%, and the rate of carbapenemase-producing Gram-negative rods was 0.5% among glucose non-fermenting Gram-negative rods and 0.3% among Enterobacteriaceae spp. The annual isolation rates of carbapenemase-producing Enterobacteriaceae (CPE) and carbapenem-resistant isolates in Enterobacteriaceae spp., *P. aeruginosa*, and *A. baumannii* are presented in Appendix A. The isolation rate of CRE was 0.2% (14/7383 strains). Four of the twenty-three CPE strains were classified as CRE. Although 12 CPE strains were isolated from seven different wards between April 2015 and March 2016, no apparent outbreak caused by CRE or CPE was experienced throughout the study period. The rate of isolation of carbapenem-resistant *P*. *aeruginosa* was 7.5%, and a slight increase was observed from April 2021–March 2022 (12.0%). The isolation of carbapenem-resistant *A*. *baumannii* was rare (3/321 strains, 0.9%). The resistance rates against each antibiotic with anti-*Pseudomonas* activity among Gram-negative species are presented in Appendix A. The non-susceptibility rate determined by the CLSI criteria is presented in Appendix A.

### 3.4. Correlation between the Isolation Rate of Antibiotic Resistant Organisms and Antibiotic Use

Using the quarterly data for 7 years (28 terms), the correlation between each antibiotic marker and the resistant rate against ≥2 or ≥3 antibiotic classes was evaluated (Appendix A). The AHI and modified AHI showed a significant negative correlation with the resistance rate against ≥2 (AHI: R^2^ 0.176, *p* = 0.026; modified AHI: R^2^ 0.320, *p* = 0.002) or ≥3 antibiotic classes (AHI: R^2^ 0.166, *p* = 0.031; modified AHI: R^2^ 0.302, *p* = 0.002) in glucose non-fermenting Gram-negative rods. A significant negative correlation was observed in the resistance rate against ≥2 antibiotic classes in Enterobacteriaceae spp. (AHI: R^2^ 0.208, *p* = 0.015; modified AHI: R^2^ 0.280, *p* = 0.004). By contrast, no correlation was found between the total DOT for broad-spectrum antibiotics and the antibiotic resistance rate in either of the Gram-negative rod groups (Appendix A).

Among the Gram-negative rod species, a significant negative correlation between the AHI or modified AHI and the antibiotic resistance rates was observed for *P*. *aeruginosa* and *K*. *pneumoniae* (Table 3 and Figure 2 and Figure 3). Notably, a high R^2^ was demonstrated in the negative correlation between the resistance rates against ≥2 antibiotic classes and the AHI (R^2^ 0.580, *p* < 0.001) or the modified AHI (R^2^ 0.500, *p* < 0.001) in *K*. *pneumoniae*. However, no significant correlation was found between the total DOT and the resistance rates against ≥2 and ≥3 antibiotic classes for any organism.

The time series of the resistant rate of these organisms are demonstrated in Figure 4. Although the resistant rate for *P*. *aeruginosa* against ≥2 classes of antibiotics was less than 15% between 4Q/2015 and 1Q/2018, the resistant rate reached 20% several times after 2Q/2018. The resistance rate for *K*. *pneumoniae* against ≥2 classes of antibiotics was approximately 5% until 3Q/2019, and, in general, the rate increased steadily thereafter. There was a tendency toward an increase in the antibiotic resistant rates in these organisms during the COVID-19 pandemic.

In the analysis for the ROC curve, significant cutoffs of antibiotic heterogeneity indices that discriminate the isolation of resistant strains were not determined for *P*. *aeruginosa*. By contrast, the significant cutoff values were 0.756 for the AHI (AUC = 0.636, *p* < 0.001) and 0.889 for the modified AHI (AUC = 0.616, *p* < 0.001) for resistant strains against ≥2 antibiotic classes in *K*. *pneumoniae*. However, the high accuracy of these antibiotic heterogeneity indices cannot be expected to discriminate the risk for resistant strains because of the low-level AUC.

The ROC curve that demonstrated the detection ability of resistant *K*. *pneumoniae* against ≥2 classes of antibiotics in the AHI, modified AHI, and overall DOT for broad-spectrum antibiotic classes is shown in Appendix A. There was no significant difference in the AUC between the AHI and the modified AHI (*p* = 0.206). 

## 4. Discussion

Determining the changes in antibiotic administration and the likely effect on resistance levels at hospitals is important to identify the priorities in terms of intervention. Veličković-Radovanović et al. [33] reported a significant negative association between cefepime use and susceptibility in multidrug-resistant *Klebsiella* sp. Troughton et al. [34] demonstrated an inverse correlation between the use of ciprofloxacin and susceptibility to ciprofloxacin among Gram-negative organisms. In addition, reducing fluoroquinolone exposure exerted a favorable effect on not only fluoroquinolone but also β-lactam and gentamicin susceptibility. Ortiz-Brizuela et al. [35] found that an increase in piperacillin/tazobactam use of one defined daily dose per 100 patient days would lead to an increase of 0.69 in carbapenem-non-susceptible Enterobacteriaceae spp. isolation/10,000 patient days. However, the decrease in antibiotic use with the introduction of the AS program did not necessarily improve antibiotic susceptibility when pre- and post-intervention periods were compared. Wang et al. [7] reported that the reduced antibiotic use achieved in the AS program partly correlated with AMR control. However, time series analysis demonstrated that the resistance rates of *E*. *coli* and *K*. *pneumoniae* against carbapenems increased during the implementation of the AS program. 

Plüss-Suard et al. [11] reported that the diversity of antibiotics used in hospitals might affect not only the rate of *P*. *aeruginosa* resistance to carbapenems but also the rate of three anti-pseudomonal agents, including carbapenems. They observed a negative correlation between resistance and the diversity of antibiotic use, as measured by the Peterson index. Sandiumenge et al. [36] reported that the balanced use of different broad-spectrum antibiotics was necessary to reduce the selection pressure that promotes the development of resistance. High homogeneity was associated with an increase in carbapenem-resistant *A*. *baumannii* and ESBL-producing Enterobacteriaceae. Interestingly, compared with other strategies including antibiotic cycling, greater antimicrobial diversity was achieved following treatment by well-trained physicians with patient-specific empirical antimicrobial selection considering the length of hospitalization and recent antibiotic exposure. 

We previously developed a strategy called periodic monitoring and supervision (PAMS). [25] Recommended, restricted, and off-supervised broad-spectrum antibiotic classes against Gram-negative organisms were changed every 3 months according to the rate of use of these antibiotics. If an antibiotic class had been used infrequently (frequently) during the previous 3 months, use of that class was encouraged (discouraged) for the following 3 months. With the introduction of PAMS in September 2006, a steep rise in the AHI was obtained, and the target value was achieved after 6 months. Concomitant with successful hospital-wide diverse antibiotic use, the rates of multidrug-resistant *P*. *aeruginosa* and *A*. *baumannii* decreased to less than one third that of the baseline rate. In *P*. *aeruginosa*, the resistance rates to cefepime and piperacillin/tazobactam decreased soon after the start of PAMS, and the resistance rate to imipenem decreased 6 months after the start of PAMS.

Because high-level antibiotic heterogeneity was achieved and the restriction of carbapenems or piperacillin/tazobactam was not required between September 2007 and March 2008, PAMS was discontinued in June 2008 in our institution. Although the restriction of piperacillin/tazobactam was announced in the hospital between November 2017 and March 2018 because of a sustained high % DOT for this antibiotic, no strict strategic control of antibiotic use was implemented, and prospective audit and supervision and consultation services with frequent timeouts for the appropriate antibiotics in each patient have been conducted. In the present study, the 7-year study period was divided into 28 terms, and the quarterly overall DOT for broad-spectrum antibiotic classes and two antibiotic heterogeneity indices were compared with the resistance rate against ≥2 and ≥3 anti-*Pseudomonas* antibiotics. The total broad-spectrum antibiotic use was not a significant factor in increasing the resistance rates, but a decrease in the antibiotic heterogeneity indices caused an increase in the resistance rates for *P*. *aeruginosa* and *K*. *pneumoniae*. 

Based on the DOT % for each broad-spectrum antibiotic derived from our 7-year time series, a rate of 25% use of fluoroquinolones, which is the target for the AHI (the index for the evaluation of equal use of four broad-spectrum antibiotic classes), might be too high to achieve in clinical practice in Japan. Hence, we proposed a modified AHI specifying 10% use of fluoroquinolones and 30% use of each of the remaining three antibiotic classes. A higher value was achieved using the modified AHI compared with the original AHI. The cutoff values that discriminated the isolation of resistant *K*. *pneumoniae* against ≥2 tested antibiotics were 0.756 in the AHI and 0.889 in the modified AHI, respectively. Although the modified AHI appears to be a feasible index in Japan, there was no significant difference in the AUC between the original AHI (0.636) and the modified AHI (0.616). If the discrimination ability for the increased risk of antibiotic resistance is 100%, the AUC is 1, whereas if the discrimination ability is 0%, the AUC is 0.5. Thus, we were unable to conclude which was the better index for balanced antibiotic use to decrease the risk of the isolation of resistant Gram-negative rods.

In previous studies, considerable differences in AMR rates have been reported for different countries [37,38,39,40]. To provide an unbiased evaluation for our study, the AMR rates of *P*. *aeruginosa* and *K*. *pneumoniae* were compared with the AMR rates derived from Japanese nationwide surveillance [41,42]. The rates of non-susceptible *P*. *aeruginosa* isolated from patients with respiratory tract infections [41] were 17.6% against meropenem, 20.1% against piperacillin/tazobactam, 15.1% against cefepime, 17.0% against ciprofloxacin, and 2.5% against tobramycin, according to the CLSI criteria. The rates of non-susceptible *K*. *pneumoniae* isolated from postoperative intra-abdominal infections [42] were 8.9%, 9.4%, 4.7%, 9.4%, and 1.6% [gentamicin], respectively. In the present study, similar non-susceptibility rates were demonstrated in *P*. *aeruginosa* (16.7% against meropenem, 13.1% against piperacillin/tazobactam, 16.1% against cefepime, 15.0% against ciprofloxacin, and 17.6% against gentamicin or amikacin). However, variable differences between the non-susceptibility rates in this study and that of nationwide surveillance were found in *K*. *pneumoniae* (the present study: 1.1% in meropenem%, 3.9% in piperacillin/tazobactam, 9.8% in cefepime, 6.7% in levofloxacin, and 6.2% in gentamycin or amikacin). 

This study has several limitations. First, the results obtained from the day-to-day activity of the AS program were analyzed retrospectively. Second, the COVID-19 pandemic may have impaired the activities of the AS program [43,44] and may have caused an increase in the AMR rate [45,46]. All members of the AS team were involved in the hospital management of COVID-19. Reduced antibiotic diversity was demonstrated from 4Q in 2020, and increased rates of resistant *P*. *aeruginosa* and *K*. *pneumoniae* were observed from 2Q in 2020, both of which had a substantial effect on the results of this study. The annual number of interventions or supervisions has been sustained during the pandemic. However, a reduced adherence rate for intervention might decrease the antibiotic diversity, but this rate was not available. The increased % DOT with carbapenems should have been recognized and corrected earlier. Third, although the resistance rate for *K*. *pneumoniae* had a substantial impact on the results, only seven of the isolated strains of this organism produced carbapenemase. Fattouh et al. [47] reported that 14% of CPE isolates were not captured by the CLSI screening breakpoint (MIC ≥2 μg/mL). Carbapenemase-producing *K*. *pneumoniae* from clinical isolates were found to produce IMP-type carbapenemase in Japan. Furthermore, 83 of 104 CPE isolates were IMP-6 producers, the majority of which simultaneously produced CTX-M type ESBL. Meropenem MIC <2 μg/mL rates were 75.0% in IMP-6-positive *K*. *pneumoniae*, suggesting that several IMP-6-producers might not have been identified by routine screening [48]. Therefore, a revised laboratory screening method for CPE may be needed, especially in Japan. Finally, in addition to antibiotic use, infection control practices also have a significant effect on the isolation of AMR organisms.

We propose the following measures for the balanced use of broad-spectrum antibiotics to decrease the AMR based on the results. First, the establishment of a comprehensive AS program, including education and a prospective audit and feedback, aimed at achieving appropriate antibiotic use in each patient is required. There is a tendency for the prolonged use of broad-spectrum antibiotics, such as carbapenems and piperacillin/tazobactam, which causes selection of AMR organisms with resulting poor susceptibility to these antibiotics. Hence, planned changes to other antibiotics should be considered within 7 to 10 days based on the culture results and the patient’s clinical course. Second, even with appropriate antibiotic selection or duration of administration in each individual patient, hospital-wide antibiotic use may also become unbalanced over time. Therefore, the early detection of the excess use of particular antibiotics by periodic antibiotic monitoring is required to optimize hospital-wide antibiotic use. Third, from our experience, substantial full-time equivalents for AS activity of the core members should be sustained during the ongoing COVID-19 pandemic because AS activity might be impaired as a result of the additional roles gained by staff during this time (such as the supervision of therapy with newly introduced medicines for COVID-19, vaccination roll-out, the updating of quarantine and isolation rules in hospitals, and the management of infection clusters).

## 5. Conclusions

The balanced use of broad-spectrum antibiotics and antibiotic diversity are more important in preventing AMR among Gram-negative rods than the total use of broad-spectrum antibiotics. This study underscores the importance of the routine monitoring of antibiotic heterogeneity indices, which are easy to calculate as a convenient tool for the AS metric. Further studies are required to determine the ideal ratio of broad-spectrum antibiotic classes for diversified use to prevent AMR. 

## Figures and Tables

**Figure 1 pharmaceutics-15-00518-f001:**
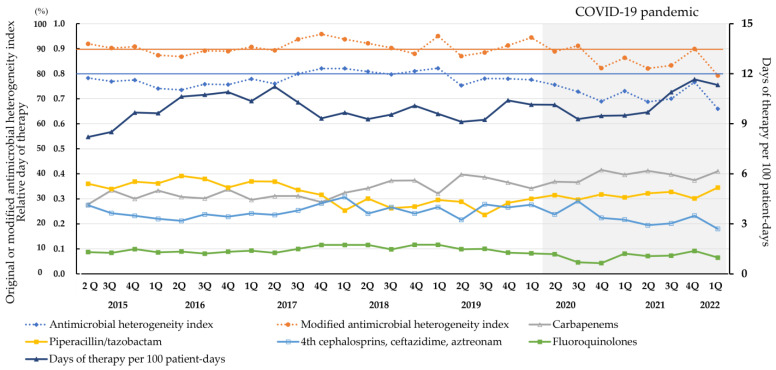
Quarterly change in the antimicrobial heterogeneity index (AHI), the modified AHI, the days of therapy per 100 patient days, and the relative days of therapy for broad-spectrum antibiotics predominantly used for hospital-onset infections.

**Figure 2 pharmaceutics-15-00518-f002:**
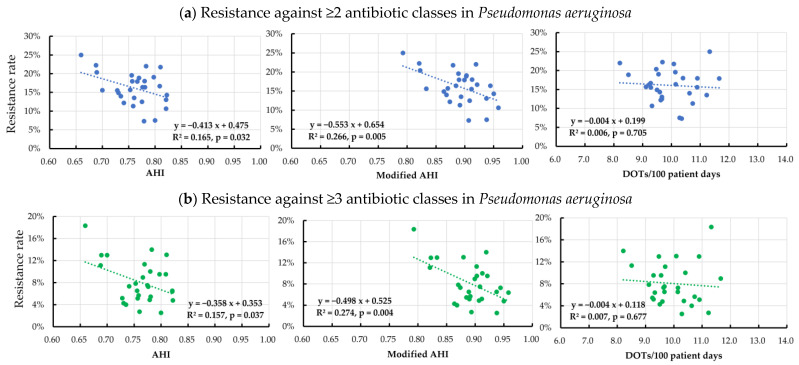
Correlation between the quarterly resistance rate in *Pseudomonas aeruginosa* and the quarterly antimicrobial heterogeneity index (AHI), modified AHI, and days of therapy per 100 patient days for broad-spectrum antibiotics predominantly used for hospital-onset infections between April 2015 and March 2022.

**Figure 3 pharmaceutics-15-00518-f003:**
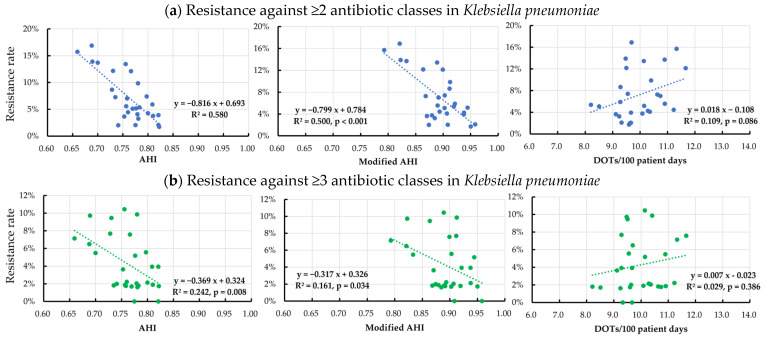
Correlation between the quarterly resistance rate in *Klebsiella pneumoniae* and the quarterly antimicrobial heterogeneity index (AHI), modified AHI, and days of therapy per 100 patient days for broad-spectrum antibiotics predominantly used for hospital-onset infections between April 2015 and March 2022.

**Figure 4 pharmaceutics-15-00518-f004:**
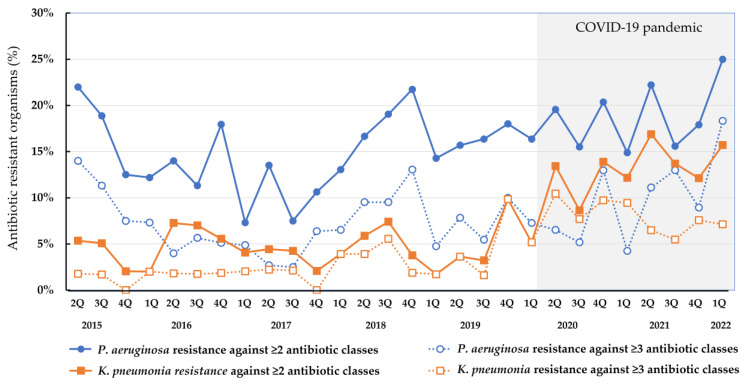
Quarterly change in the resistance rate in *Pseudomonas aeruginosa* and *Klebsiella pneumoniae* between April 2015 and March 2022.

**Table 1 pharmaceutics-15-00518-t001:** Annual number of events according to the reasons for the intervention/supervision between April 2015 and March 2022.

Reasons for the Initial Intervention/Supervision	2015 April–2016 March	2016 April–2017 March	2017 April–2018 March	2018 April–2019 March	2019 April–2020 March	2020 April–2021 March	2021 April–2022 March	Total
Infectious disease consultation service	684 (69.2%)	657 (65.0%)	613 (54.9%)	602 (51.7%)	659 (56.2%)	552 (48.6%)	668 (54.6%)	4435 (56.9%)
Suggestion of antimicrobial selection and dosing for patients admitted to the intensive care unit	152 (15.4%)	158 (15.9%)	179 (16.0%)	164 (14.1%)	165 (14.1%)	172 (15.2%)	166 (13.6%)	1156 (14.8%)
Isolation of organisms from the aseptic specimen	92 (9.3%)	97 (9.8%)	97 (8.7%)	94 (8.1%)	95 (8.1%)	143 (12.6%)	135 (11.0%)	753 (9.7%)
Isolation of antimicrobial-resistant organisms	9 (0.9%)	18 (1.9%)	31 (2.8%)	51 (4.4%)	51 (4.4%)	52 (4.6%)	51 (4.2%)	263 (3.4%)
Diagnosis of *Clostridioides difficile* infection	42 (4.2%)	31 (3.1%)	37 (3.3%)	33 (2.8%)	34 (2.9%)	47 (4.1%)	30 (2.5%)	254 (3.3%)
Prescription for carbapenems or piperacillin/tazobactam *	–	–	87 (7.8%)	144 (12.4%)	92 (7.8%)	121 (10.7%)	104 (8.5%)	548 (7.0%)
Prolonged use of designated antimicrobial agents	10 (1.0%)	35 (3.6%)	73 (6.5%)	76 (6.5%)	76 (6.5%)	48 (4.2%)	69 (5.6%)	387 (5.0%)
Total	989	996	1117	1164	1172	1135	1223	7796

* Prospective audit and feedback were started since November 2017 for piperacillin/tazobactam and September 2019 for carbapenems.

**Table 2 pharmaceutics-15-00518-t002:** Isolation of resistant organisms among glucose non-fermenting Gram-negative rods and Enterobacteriaceae spp.

Organisms	No. of Strains	No. of Resistant Organisms (Rate)		Extended-Spectrum β-Lactamase-producing Organisms	Carbapenemase-Producing Organisms
		≥2 Classes	≥3 Classes		
Glucose non-fermenting Gram-negative rod	1713	248 (14.5%)	123 (7.2%)	–	8 (0.5%)
*Pseudomonas aeruginosa*	1392	228 (16.4%)	117 (8.4%)	–	8 (0.6%)
*Acinetobacter baumannii*	321	20 (6.2%)	6 (1.9%)	–	0 (0.0%)
Enterobacteriaceae sp.	7838	1102 (14.1%)	297 (3.8%)	897 (11.4%)	23 (0.3%)
*Escherichia coli*	2863	765 (26.7%)	190 (6.6%)	702 (24.3%)	3 (0.1%)
*Klebsiella pneumoniae*	1685	131 (7.8%)	79 (4.7%)	155 (8.8%)	7 (0.4%)
*Klebsiella oxytoca*	670	26 (3.9%)	7 (1.0%)	11 (1.5%)	0 (0.0%)
*Klebsiella aerogenes*	339	16 (4.7%)	5 (1.5%)	2 (0.6%)	0 (0.0%)
*Enterobacter cloacae*	960	123 (12.8%)	10 (1.0%)	9 (0.8%)	9 (0.9%)
Other Enterobacteriaceae sp.	1321	41 (3.1%)	6 (0.5%)	18 (1.2%)	4 (0.3%)
*Serratia marcescens*	382	4 (1.0%)	1 (0.3%)	0 (0.0%)	0 (0.0%)
*Citrobacter freundii*	300	12 (4.0%)	4 (1.3%)	6 (1.7%)	3 (1.0%)
*Proteus mirabilis*	204	14 (6.9%)	0 (0.0%)	12 (5.1%)	0 (0.0%)
*Proteus vulgaris*	140	4 (2.9%)	0 (0.0%)	0 (0.0%)	0 (0.0%)
*Morganella morganii*	295	7 (2.4%)	1 (0.3%)	0 (0.0%)	1 (0.3%)

**Table 3 pharmaceutics-15-00518-t003:** Correlation between the quarterly resistance rates for each Gram-negative rod organism and the quarterly antimicrobial heterogeneity index (AHI), modified AHI, and days of therapy per 100 patient days for the broad-spectrum antibiotics predominantly used for hospital-onset infections.

Resistant Organisms	Antimicrobial Heterogeneity Index				Modified Antimicrobial Heterogeneity Index				Days of Therapy per 100 Patient Days			
	Gradient	R^2^	*p*-Value	95% CI	Gradient	R^2^	*p* Value	95% CI	Gradient	R^2^	*p*-Value	95% CI
*Pseudomonas aeruginosa*												
≥ 2 class	−0.413	0.165	0.032	−0.676 to −0.039	−0.553	0.266	0.005	−0.746 to −0.177	−0.004	0.006	0.705	−0.436 to 0.307
≥ 3 class	−0.358	0.157	0.037	−0.670 to −0.027	−0.498	0.274	0.004	−0.750 to −0.186	−0.004	0.007	0.677	−0.442 to 0.300
*Acinetobacter baumannii*												
≥ 2 class	−0.077	0.002	0.810	−0.413 to 0.331	−0.097	0.003	0.774	−0.421 to 0.323	0.020	0.061	0.204	−0.139 to 0.568
≥ 3 class	0.024	0.001	0.899	−0.351 to 0.395	0.044	0.002	0.824	−0.335 to 0.410	0.007	0.022	0.447	−0.236 to 0.495
*Escherichia coli*												
≥ 2 class	−0.079	0.006	0.697	−0.438 to −0.305	−0.199	0.034	0.350	−0.521 to 0.204	−0.009	0.032	0.363	−0.517 to 0.208
≥ 3 class	0.153	0.055	0.231	−0.152 to 0.558	0.087	0.016	0.521	−0.259 to 0.477	−0.005	0.024	0.432	−0.499 to 0.232
*Klebsiella pneumoniae*												
≥ 2 class	−0.816	0.580	<0.001	−0.894 to −0.576	−0.799	0.500	<0.001	−0.855 to−0.454	0.018	0.109	0.086	−0.048 to 0.626
≥ 3 class	−0.369	0.242	0.008	−0.765 to −0.219	−0.317	0.161	0.034	−0.673 to−0.033	0.007	0.029	0.386	−0.217 to 0.511
*Enterobacter cloacae*												
≥ 2 class	−0.338	0.047	0.268	−0.546 to 0.170	−0.188	0.013	0.562	−0.468 to 0.270	−0.014	0.032	0.364	−0.517 to 0.209
≥ 3 class	−0.084	0.038	0.320	−0.530 to 0.192	−0.087	0.037	0.330	−0.527 to 0.196	−0.002	0.007	0.676	−0.442 to 0.300

95% CI: 95% confidence interval.

## Data Availability

The data presented in this study are available from the corresponding author on reasonable request.

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
