# Peer review of "Correlation between Antimicrobial Resistance and the Hospital-Wide Diverse Use of Broad-Spectrum Antibiotics by the Antimicrobial Stewardship Program in Japan"

_pharmaceutics, 2023, doi:10.3390/pharmaceutics15020518_

Round 1

Reviewer 1 Report

Dear Editor, Dear Authors!

This study emphasizes that a balanced, varied antibiotic use of broad-spectrum antibiotics and the importance of routine monitoring of antibiotic heterogeneity indices, can be easily calculated as a convenient tool for AS measurement. As the authors rightly point out, further studies are needed to determine the ideal ratio of broad-spectrum antibiotic classes for diversified use to prevent AMR. The limiting factors also mentioned in the study, such as the effect of the reduced adherence rate for the intervention, do not, in my view, particularly bias the results. Indeed, the pandemic has affected almost all areas of medicine. Unfortunately, this phenomenon is beyond our scope, which does not detract from the authors' merit. Overall, I consider the study comprehensive, significant, and relevant. The figures and tables are adequate. The conclusions felt appropriate. Overall, the study is, in my opinion, complete and will help the development of the area. 

Sincerely,

Reviewer

Author Response

Thank you for your comments for our manuscript.

Reviewer 2 Report

Dear editor and authors,

The study is with highly importance in its field and the manuscript is well written and structured and I encourage its publication. However in some parts is hard to follow it. I suggest to the authors to add an abbreviation list in which to define all the index used, and may be the method of calculation. Although the authors stated in the conclusion that further research is needed for establishing the ideal ratio, I would encourage the authors to add in the end of the discussions some clear proposed measures to decrease the AMR based on their results. 

Author Response

  1. We added abbreviation list in Table S1 and method of calculation for antibiotic heterogeneity indices in Table S2.
  2. Line 404-421: We propose the following measures for the balanced use of broad-spectrum antibiotics to decrease the AMR based on the results. First, the establishment of a comprehensive AS program including education and a prospective audit and feedback, aimed at achieving appropriate antibiotic use in each patient is required. There is a tendency for the prolonged use of broad-spectrum antibiotics, such as carbapenems and piperacillin/tazobactam, which causes selection of AMR organisms with resulting poor susceptibility to these antibiotics. Hence, planned changes to other antibiotics should be considered within 7 to 10 days based on the culture results and the patient’s clinical course. Second, even with appropriate antibiotic selection or duration of administration in each individual patient, hospital-wide antibiotic use may also become unbalanced over time. Therefore, the early detection of excess use of particular antibiotics by periodic antibiotic monitoring is required to optimize hospital-wide antibiotic use. Third, from our experience, substantial full-time equivalents for AS activity of the core members should be sustained during the ongoing COVID-19 pandemic because AS activity might be impaired as a result of the additional roles gained by staff during this time (such as the supervision of therapy with newly introduced medicines for COVID-19, vaccination roll-out, the updating of quarantine and isolation rules in hospitals, and the management of infection clusters).

Please see the added supplementary Table prepared under the column

Reviewer 3 Report

Dear all,

The manuscript entitled “Correlation of antimicrobial resistance and hospital-wide diverse use of broad-spectrum antibiotics by antimicrobial stewardship program in Japan“ is overall well written and reports important data on antibiotic use in terms of quality vs. quantity, as well as its relationship with the emergence of multidrug-resistant Gram-negative strains in the clinical setting. Even though the data presented is local and not extrapolable to other regions of the globe, it is nonetheless an important piece of evidence on the trend followed by microorganisms with regard to antimicrobial non-susceptibility. Thus, I believe the manuscript should be accepted after minor revisions.

Some comments:

  1. The fact that the present study is focused on Gram-negative strains is of extreme importance given the current scenario of global concern with this type of bacteria (e.g. due to increased propensity for antimicrobial resistance development, overall lower antibiotic susceptibility, and higher death rates due to AMR). However, the authors do not sufficiently explore this aspect in their introduction or discussion. I suggest the authors do so by citing, for instance, the latest systematic analysis published in the Lancet Journal by the Antimicrobial Resistance Collaborators earlier this year (https://doi.org/10.1016/S0140-6736(21)02724-0). Other references highlighting CRAB, CRAP and CRE as the top priority microorganisms for which new antibiotics are urgently needed, as established by the WHO, are also welcome.

  2. The rate of carbapenem-resistant Gram-negative isolates should also be highlighted in the context of the WHO priority microorganisms.

  3. Concepts such as AHI, modified AHI and ROC could be explained further for readers who are not entirely familiar with such terms. This would facilitate data interpretation, particularly with regard to the practical implications of the data shown in the manuscript.

  4. Lines 23-26: The sentence is too long and a bit confusing, please rephrase.

  5. Line 30: Please avoid using “etc”. Rather, specify the antibiotics covered by the analysis.

  6. Line 43: “Prior use of antibiotics was…”: Please either change “was” to “is” or indicate clearly that this comes from a previous study.

  7. Line 56: It seems that something is missing from this sentence. I suggest changing it to: “... we proposed the employment/application/use of the antibiotic heterogeneity index” if this is the idea that the authors indeed wish to convey.

  8. Line 57: Please consider replacing “and increased AHI resulted” with “and have observed that an increased AHI”.

  9. Line 157: The definition of “event” must be clearly stated before the mean annual number of events is presented (e.g. In this study an event was defined as…”).

  10. Line 185: “..AHI is an easier antibiotic heterogeneity marker…” Easier than what? Please state the comparison reference.

  11. Lines 219-224: When describing correlation significance, please state p-values and R2 values throughout the text between brackets to facilitate data interpretation. 

  12. Line 318: Please state the reason why (if known) PAMS was discontinued.

  13. Line 336-337: The practical implications of the trend noticed in this sentence should be further discussed. Otherwise, the reader may not capture the real importance of these results.

  14. Line 341: “A considerable difference in AMR rates was found…” Please replace “was” with “has been”. Also, please start the sentence with: “In previous studies”.

  15. Line 365: Please remove the comma after “although”.

Kind regards.

Author Response

  1. Line 41-48: Predictive statistical models have estimated that 4.95 million deaths annually are associated with AMR, and the six leading pathogens, namely Escherichia coli, Staphylococcus aureus, Klebsiella pneumoniae, Streptococcus pneumoniae, Acinetobacter baumannii, and Pseudomonas aeruginosa were responsible for 3.57 million deaths associated with AMR in 2019 [3]. Carbapenem-resistant Gram-negative bacteria, including carbapenem-resistant Enterobacteriaceae (CRE), baumannii, and P. aeruginosa cause difficult to treat infections and are associated with a poor prognosis [4].
  2. Line 228-236: The annual isolation rates of carbapenemase-producing Enterobacteriaceae (CPE) and carbapenem-resistant isolates in Enterobacteriaceae spp., Pseudomonas aeruginosa, and Acinetobacter baumannii are presented in Table S6. The isolation rate of CRE was 0.2% (14/7383 strains). Four of the 23 CPE strains were classified as CRE. Although 12 CPE strains were isolated from seven different wards between April 2015 and March 2016, no apparent outbreak caused by CRE or CPE was experienced throughout the study period. The rate of isolation of carbapenem-resistant aeruginosa was 7.5%, and a slight increase was observed from April 2021–March 2022 (12.0%). The isolation of carbapenem-resistant A. baumannii was quite rare (3/321 strains, 0.9%).

Table S6. Annual change in the isolation rates of carbapenemase-producing Enterobacteriaceae and carbapenem-resistant isolates in Enterobacteriaceae sp., Pseudomonas aeruginosa, and Acinetobacter baumannii

  1. As explanation of the method for AHI and modified AHI, Table S2 was added.

Line 353-357: Based on the % DOT for each broad-spectrum antibiotic derived from our 7-year time series, a rate of 25% use of fluoroquinolones, which is the target for the AHI (index for the evaluation of equal use of four broad-spectrum antibiotic classes) might be too high to achieve in clinical practice in Japan. Hence, we proposed a modified AHI specifying 10% use of fluoroquinolones and 30% use of each of the remaining three antibiotic classes.

Line 362-364: If the discrimination ability for the increased risk of antibiotic resistance is 100%, the AUC is 1, whereas if the discrimination ability is 0%, the AUC is 0.5.

  1. Line 23-26: The days of therapy/100 patient-days (DOT) for four broad-spectrum antibiotic classes were evaluated for 2015– The antibiotic heterogeneity index (AHI) for the equal use of four classes (25%) and the modified AHI for the equal use of three classes (30%), excluding fluoroquinolones (10%), were measured (target: 1.0).
  2. Line 28-29: 24.3% for fourth-generation cephalosporins/ceftazidime/aztreonam
  3. Line 52-53: Prior use of antibiotics is the most significant risk factor for the acquisition of resistant Gram-negative organisms
  4. Line 64-65: Previously, we proposed the employment of the antibiotic heterogeneity index (AHI) for common broad-spectrum antibiotics with anti-Pseudomonas activity
  5. Line 65-66: and have observed that an increased AHI resulted in a decrease in the isolation rate of AMR organisms
  6. Line 169-170: Because there was more than one episode at which AS could be initiated in some patients, the number of events was evaluated instead of the number of patients. Line 182-183: The annual number of events according to the reasons for the intervention/supervision is presented in Table 1.
  1. Line 199-202: The AHI and modified AHI were calculated as 0.775 and 0.909, respectively. As 1.0 was the target for both indices, the modified AHI is a more feasible antibiotic heterogeneity marker to achieve a value closer to the target than the AHI in Japanese hospitals.
  2. Line 246-258: The AHI and modified AHI showed a significant negative correlation with the resistance rate against ≥2 (AHI: R2176, p = 0.026; modified AHI: R2 0.320, p = 0.002) or ≥3 antibiotic classes (AHI: R2 0.166, p = 0.031; modified AHI: R2 0.302, p = 0.002) in glucose non-fermenting Gram-negative rods. A significant negative correlation was observed in the resistance rate against ≥2 antibiotic classes in Enterobacteriaceae spp. (AHI: R2 0.208, p = 0.015; modified AHI: R2 0.280, p = 0.004). By contrast, no correlation was found between the total DOT for broad-spectrum antibiotics and the antibiotic resistance rate in either of the Gram-negative rod groups (Table S9). Among the Gram-negative rod species, a significant negative correlation between the AHI or modified AHI and the antibiotic resistance rates was observed for P. aeruginosa and K. pneumoniae (Table 3 and Figures 2, 3). Notably, a high R2 was demonstrated in the correlation between the resistance rates against ≥2 antibiotic classes and the AHI (R2 0.580, p < 0.001) or the modified AHI (R2 0.500, p < 0.001) in K. pneumoniae.
  1. Line 339-341: Because high-level antibiotic heterogeneity was achieved and restriction of carbapenems or piperacillin/tazobactam was not required between September 2007 and March 2008, PAMS was discontinued in June 2008 in our institution.
  2. Line 353-366: Based on the % DOT for each broad-spectrum antibiotic derived from our 7-year time series, a rate of 25% use of fluoroquinolones, which is the target for the AHI (index for the evaluation of equal use of four broad-spectrum antibiotic classes) might be too high to achieve in clinical practice in Japan. Hence, we proposed a modified AHI specifying 10% use of fluoroquinolones and 30% use of each of the remaining three antibiotic classes. A higher value was achieved using the modified AHI compared with the original AHI. The cutoff values that discriminated the isolation of resistant pneumoniae against ≥2 tested antibiotics were 0.756 in the AHI and 0.889 in the modified AHI, respectively. Although the modified AHI appears to be a feasible index in Japan, there was no significant difference in the AUC between the original AHI (0.636) and the modified AHI (0.616). If the discrimination ability for the increased risk of antibiotic resistance is 100%, the AUC is 1, whereas if the discrimination ability is 0%, the AUC is 0.5. Thus, we were unable to conclude which was the better index for balanced antibiotic use to decrease the risk of isolation of resistant Gram-negative rods.
  3. Line 367-368: In previous studies, considerable differences in AMR rates have been reported for different countries
  4. Line 392-393: Third, although the resistance rate for pneumoniae

Please see the added supplementary Table prepared under the column.

Reviewer 4 Report

Manuscript revision for Pharmaceutics-2088965

Comments to authors:

In this manuscript, the authors present an experience of “innovative” indices to monitor and to feed-back prescribers about the use of four major antimicrobial classes.

These indices aim at evaluation whether the main use of one class or rather than the variability of prescription among classes may impact the onset or the progression of AMR rate among the most dreadful gram-negative pathogens.

The paper is well-written, and sections are well built.

There are very few mistakes or misspellings, which can be quickly corrected, such as “Klebsiella pneumonia” instead of “pneumoniae”, or the reference n. 30, which presents a misspelling at line 282.

However, this experience shows a major flaw that may have an important impact on both overall and specific results presented (see paragraph 3.3): the authors used the CLSI interpretation of antibiograms, which splits results in three major definitions; S=susceptible, I=intermediate susceptibility, and R=resistance. EUCAST (European Committee for Antimicrobial Susceptibility Testing) has re-defined those three definitions, where “S” and “R” remained as such, but “I” no longer means Intermediate Susceptibility, that is with a borderline profile between susceptible and resistant, which always falls into the Resistance rate and not into the Susceptible one. Rather, nowadays “I” means susceptible Increased or Improved dose: if this concept implies that the strain isolated remain fully susceptible to an antimicrobial drug, provided that the its dosing has been assessed and informed on PK/PD indices and recommendations.

This also means that epidemiological results of AMR rates, especially among gram-negative important pathogens, should be revised accordingly, thus not only recruiting much more susceptible strains on the one side, with complete different results on antimicrobial usage and threats; but also allowing comparison to and matches with other experiences already using such new criteria on the other side.

This fact hampers the experience and results may not be reliable to draw conclusions on the efficacy and effectiveness of the parameters used.

Author Response

  1. Misspelling for Klebsiella pneumoniae, or the reference n. 30 were amended.
  2. We reevaluated the antibiotic susceptibility based on the EUCAST (Table 2, 3 and Figure 4) and demonstrated the correlation between the resistant rate (EUCAST) and antibiotic use markers (Figure 2, 3) Line 138-142: Antibiotic susceptibility was determined according to the European committee on antimi-crobial susceptibility testing (EUCAST) and the Clinical and Laboratory Standards Insti-tute (CLSI) criteria [29] (Table S3). As the primary endpoint for antibiotic susceptibility, re-sistance in EUCAST against ≥2 or ≥3 of the following antibiotic classes was analyzed.
  3. MIC breakpoints based on EUCAST and the CLSI in Pseudomonas aeruginosa, Acinetobacter baumannii, and Enterobacteriaceae spp.was demonstrated in Table S3.

Please see the added supplementary Table prepared under the column

Reviewer 5 Report

The article is interesting, well articulated, the results well explained and conclusions well discussed.

The differences that occurred due to COVID-19 are reported, I would suggest inserting a space or a line in the graphs that underline the Covid period to highlight the differences that occurred in this period.

In Discussion: you said that "several IMP-6-producers might have been escaped from routine screening". I agree with your idea, in our laboratories we are now at IND-16!

Just a small fix: line 300, please replace brad with broad.

Author Response

  1. According to reviewer’s comment, we inserted COVID-19 pandemic period in Figure 1 and 4. Line 211-213: However, a decrease in both antibiotic heterogeneity indices was observed from 4Q/2020 onwards, when more than 2000 new daily cases of COVID-19 were being reported for the first time in Japan. Line 283-285: There was a tendency toward the increase of the antibiotic resistant rates in these organisms during the COVID-19 pandemic.
  2. Line 320: different broad-spectrum

Please see the added supplementary Table prepared under the column

Round 2

Reviewer 4 Report

I think the authors have answered to all questions posed; the manuscript has therefore been improved and reached soundness for readers.